# Local School Desegregation Practices in Sweden

Ali Osman * and Stefan Lund 

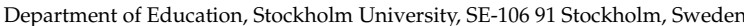

Department of Education, Stockholm University, SE-106 91 Stockholm, Sweden
* Correspondence: ali.osman@edu.su.se

**Abstract:** The focus of this paper is to examine the desegregation of hyper-segregated schools in Sweden. In this paper, our focus is on the practice(s) of desegregation in four Swedish municipalities. This study is based on interviews with municipal school politicians, school headmasters, and municipal school officials. Theoretically, the study departs from the theory of steering. This study shows that the municipalities use a strong belief in the peer effect to legitimise their decisions to desegregate hyper-segregated schools. Their decisions ignore a substantial research body that could lead to the development of different types of pedagogies that are relevant for these types of schools.

**Keywords:** school integration; migration

## 1. Introduction

The focus of this paper is to examine the practices of four Swedish municipalities to change the composition of the student body in the compulsory school system. The target of the practice are schools with a high composition of students with low socioeconomic status (SES) and a migrant background. These schools are often labelled as "hyper-segregated" schools and are located in socially stigmatised residential areas. Hyper-segregated schools in this paper are defined as schools in which a large majority of the student body is of minority background, and such schools perform way below average in relation to other schools in the municipality [1]. In the Swedish case, these types of schools primarily consist of children with an immigrant background. The category "immigrant background" refers to children born in another country or whose parents were born outside of Sweden. In 2020, 26% of Sweden's population was either foreign-born or had two parents born abroad [2]. The policy actors in this paper are not talking about children with Scandinavian or West European immigrant backgrounds, or undocumented immigrant children. Instead, the focus is on children who are visibly identifiable as not Swedish, speak accented Swedish, and/or are associated with tough, immigrant neighbourhoods and thuggish behaviour [3].

In Sweden, like most countries, there is an ongoing academic and political debate regarding how to manage school inequality as a consequence of migration and segregation. The municipalities concerned, as evident below, have tried ameliorating the negative impact of school segregation by increasing the resources in these types of schools and putting in place different compensatory measures.

> In the last 10 years we have implemented major reforms, but these reforms and the measures we introduced did not result or were not reflected in the children's academic performance. We have through different types of incentives and financial support from the government put a lot of money and resources into these schools. (Official)

> We have seen [that] even after providing this school with different kinds of resources, their academic performance did not improve, and we concluded that we needed to radically restructure the system as a whole. However, the changes had to be economically and pedagogically viable or sustainable. (Politician)

The compensatory strategy, as evident above, did not improve the academic performance of the schools. The interviewees in this study state that the gap between students' average grades in hyper-segregated schools at those of students at the best-performing schools have instead increased. Against this background, we analyse how four municipalities use different desegregation practices to change the student composition of hyper-segregated schools. The paper addresses the following questions:

What different forms of steering practices for desegregation are the municipalities using to integrate students of migrant background into schools with a majority of ethnically Swedish students?

How are these desegregation practices implemented?

### 1.1. Brief Historical Contextualisation

The desegregation practices examined in this paper are local policies and practices that municipalities adopt. They are not mandated by the government and are not a policy at the national level. This reflects a recent shift in Swedish educational policy from centralised compensatory strategies towards decentralised local initiatives. To understand the current local trends in desegregation policy and practice in Swedish municipalities, it is important to very briefly describe the policies that have contributed to increased school segregation and educational inequality in Sweden. These are the decentralisation of the school system, the introduction of the school voucher system, and the proximity principle, along with residential segregation.

The transfer of the responsibility for education from the central government to Sweden's 290 municipalities began in 1988 [4]. Municipalities were given a single grant to finance obligatory welfare services, including the organisation of preschools, compulsory schools, and upper secondary schools. Until the beginning of the 1990s, as noted by Grannäs and Frelin [5], "state funded, municipally run schools were the norm, with very few exceptions, pupils attended the school closest to their home." During the 1990s in Sweden, like in many other capitalist welfare states, schools were steered according to neoliberal ideas and school choice, and a voucher system was introduced. The underlying idea was that creating a school market system with competition between schools would increase their performance, and poorly performing schools would be weeded out. Furthermore, competition would lead to more effective use of public funds. In other words, the "best" or "most appropriate" schools would attract students, and as a consequence, the overall performance of schools would increase [6]. In 2022, approximately 17.5% of all compulsory schools in Sweden were independent schools. No school charges are allowed, but independent schools can be profit-driven. Approximately 90% of independent schools are profit-driven and owned by corporations [7,8].

Unlike public schools (which are operated by the municipalities), independent schools are not obliged to cater for all types of children. This means that independent schools can tailor their admission policies and practices to choose and attract a certain type or category of children. In contrast, municipal or public schools do not have the luxury of picking and choosing who they admit; they are obliged by law to enrol all children in their catchment area. In addition, public schools are awarded the same financial support as independent schools, without any additional support for their obligatory social role to provide education to all children irrespective of their backgrounds. Two trends can be discerned in the new liberal turn and the reforms of the educational system in Sweden in the 1990s. The first is that economic austerity measures inspired by New Public Management shaped the decentralisation reforms of the Swedish school system and led to the introduction of publicly financed independent schools. This reform was framed as "school choice", turning the school into a quasi-market in Sweden. The second trend is that "a goal-steered governing system" was introduced, which increased the focus on "accountability and merit scores" [5]. A large number of studies in Sweden have suggested that these reforms have not benefited all children and that:

Those who benefited most from the school choice reforms were middle-class students with better school results and higher credit scores, rather than those with a lower SES background, a different ethnicity where a lack of language skills can be an obstacle and those with complex learning needs [5].

There is also strong evidence that the socioeconomic and ethnic composition of a school's student body is shaped by the population and demographic characteristics of the residential areas in which the school is embedded. In other words, residential segregation, school choice, and marketisation have amplified school segregation [9]. Grannäs and Frelin [5] similarly pointed out that school choice policy has been criticised for its inability to effectively combat "the negative effects of segregation, meaning that most municipalities fail to use already available measures, such as differentiated funding and the zoning of school uptake areas". Finally, while segregation levels in school catchment areas decrease from elementary to middle and high school, the segregations between catchment areas and the schools that serve them remain constant across all levels.

To summarise, decentralisation, school choice, and the introduction of private sector management regimes/styles and privatisation were legitimised on the grounds that they would enhance the quality and efficiency of service provision while lowering its cost. Nonetheless, as noted above, the reforms increased school inequality and created hyper-segregated schools, and the municipalities are struggling to manage or deal with the "effects" of the reforms.

*1.2. Research on School Segregation and Desegregation*

The international and Swedish literature on school segregation and desegregation is vast and cannot be covered within the format of a paper. As discussed in the introduction, school desegregation is largely a result of school segregation [10,11]. To understand segregation, Massey, White, and Phua [12] identified five aspects of segregation: evenness, exposure, concentration, centralisation, and clustering. Each is conceptually precise and relates to a different dimension of social segregation: concentration, centralisation, and clustering are spatial in nature. Concentration denotes the relative amount of physical space a specific people group are confined in. Centralisation is the degree to which members settle in and around the centre of an urban area. Clustering is the extent to which minority groups adjoin one another in a space, which is maximised when minority groups cohere in a neighbourhood to form one large territorial and stigmatised area, and is minimised when they are scattered widely in a space, as on a checkerboard. Exposure is the degree of potential contact between majority and minority members within a neighbourhood. Finally, evenness is the degree to which the percentages of minority members within individual residential areas approach the percentage of minority members of the entire urban area as a whole.

In Sweden, hyper-segregated areas and schools are characterised by their population or demographics being composed of people with different ethnicities and nationalities in specific geographic areas, such as in the major cities and towns, who are generally immigrants or of immigrant background [13,14] In other words, these Swedish territorialised, stigmatised areas have a high concentration of non-White migrants, which, in turn, affects the composition of the student body in schools. Such areas are, according to Wacquant:

> ... isolated and bounded territories increasingly perceived by both outsiders and insiders as social purgatories, leprous badlands at the heart of the postindustrial metropolis where only the refuse of society would accept to dwell ... Even the societies that have best resisted the rise of advanced marginality, like the Scandinavian countries, are affected by this phenomenon of territorial stigmatisation linked to the emergence of zones reserved for the urban outcasts [15].

The parents of children in hyper-segregated schools are predominantly immigrants "who arrived with low levels of education, who typically have ended up in low status

and low-paid jobs, and who also are stigmatized in the receiving society because of their ethnoracial backgrounds" [16]. According to Adamson and Galloway [17], residential segregation between Blacks and Whites in the USA is slightly diminishing but remains high, while it is growing for Whites vs. Latinx and Asians, which contributes to the segregation of schools.

In the United States, researchers on residential racial segregation have shown that race impacts neighbourhood choice [18]. The racial preference within racially homogeneous residential areas in the United States is rooted in the history of racial segregation. Monarrez, Kisidam, and Chingos noted that: "During the years of de jure segregation, a school was segregated as a matter of legal construction. In most cases, schools were either 100 percent black or 100 percent white" [19]. A European report on the nature of segregation in Europe described how children of refugees and migrants are often enrolled in schools with a high proportion of other migrants and children with a refugee background:

> Refugee children and children with a migrant background also experience school segregation in many European countries, as they are often taught in schools with a disproportionately high presence of other migrant children. . . . The Commissioner has noted with concern that children with a migrant background have also at times tended to be overrepresented in special education. The recent increase in the number of refugees arriving in Europe is making the issue of school segregation more acute, as many member states have yet to develop comprehensive integration policies that effectively secure the right of migrant and refugee children to mainstream education [20].

The report by the Commissioner for Human Rights identified a number of factors that are linked to school segregation: language difficulties, parental preferences, and White flight from schools with high percentages of children with a migrant background. These factors are also supported by other studies [21–23]. However, according to Taeuber and James [24,25], children or students from high-SES backgrounds always find ways to maintain their position by separating themselves from children with low SES.

International studies confirm Swedish research, particularly concerning the impact of socioeconomic status and vouchers on segregation in school. Smith and Meier [26] showed how parents, particularly in families with high cultural capital, tended to avoid schools with high numbers of students with low cultural and social capital. Waslander and Thrupp [27] showed the same process and effect in New Zealand. Further to those, a number of studies showed how racial segregation in school was due to both "White flight" and "self-segregation" along the lines of race and class [28].

As evident above, the performance of schools and children in segregated areas has been on the agenda in different countries, and Sweden is not an exception. The policy and the practices examined in this paper have been put in place to turn around and improve the academic performance of children with an immigrant background and achieve educational equity for all children irrespective of their background in the different municipalities. In this context, it is important to stress that there is no consensus in research on the impact of school desegregation on children's school performance. However, some studies show that desegregation improves the academic performance of minority children and students, without negatively impacting the academic performance of the dominant group [22,29]. Similarly, Billings et al. [30] and Delmont [31] concluded in their studies that the academic performance of minority students improved in socially and ethnically mixed schools, while others pointed out that desegregation often led to "White flight" [32,33].

These phenomena and processes are evident in the Swedish context. For instance, Kornvall and Bender [34] examined how municipalities attempt to manage school segregation. In one case, four middle schools were closed and a new school was built to accommodate the children from the four schools, but middle-class families counteracted the attempt by transferring their children to independent schools. Some municipalities attempt to steer the composition of children in schools by placing children with minority backgrounds in ethnically Swedish-dominated schools, and studies have shown that the school achieve-

ment of these students increases [35]. This increase in minority children's performance is explained by noting how they gained study-motivated peers [36]. In contrast, however, Jämte et al. [37] found that distributing minority-background children among middle-class schools had no impact on the minority children's school performance.

This short review of research in the area of segregated schools and desegregation shows mixed results. Segregated schools in stigmatised areas are a major problem that needs to be addressed. Yet, so far, research in this area does not seem to provide policymakers with a clear path and tools to deal with this complex problem.

*1.3. The Notion of Steering*

According to Theisens, Hooge, and Waslander [38], an important aspect of governance is steering: "In its most succinct form, steering can be defined as 'exercising influence', a definition that makes it abundantly clear that steering is a characteristic of relations between actors" [39] noted that "the governance turn may be defined in brief as a shift from centralised and vertical hierarchical forms of regulation to decentralised, horizontal, networked forms", and went on to say, citing Rosenau [40] that governance is understood as a continuum that "stretches between the transnational and the subnational, the macro and micro, the informal and the institutionalised, the state-centric and the multi-centric, the co-operative and the conflictual" [39]. Ozga furthermore stressed that the modern form of governance faces the problem of "how to govern without governing" [39]. It, therefore, generates a set of refined tools for steering policy such as standardisation, quality benchmarking, and harmonisation of data. On the production of:

> governing knowledge. . . . The key to this system lies in inculcating new norms and values by which external regulatory mechanisms transform the conduct of organisation and individual in the capacity as 'self-actualising' agents so as to achieve political objectives through action at a distance [39].

Reardon et al. [41] noted that in the governance literature, "this is how steering is understood in the main; with such definitions implying that structures of formal power and/or influential individual agency are important for determining outcomes" [41]. Thus, policy implementation is a premeditated act, which has the objective to effect change. However, Reardon et al. stressed that focusing on policy implementation is not adequate when thinking about steering, and noted that:

> The implementation of an individual policy is a complex process and it is therefore critical to consider the interactions of multiple policies, their layering and interaction over time. Issues such as the rising or fading of the importance of policies as external circumstances change and the creation (intended or otherwise) of new policy nexuses are not often in view [41].

The outcome or the change of a practice is not "simply the result of exogenous policy prescriptions or individual agent acting on a system, but rather evolve[s] endogenously over time along with changing social meanings, technologies, resources and competencies" [41]. In this paper, our objective is to describe the steering practice(s) municipalities are using to decide school desegregation policies. Our analytical focus is directed towards the ways in which the policymakers express what they are doing.

## 2. Methods

The data collection for this study was conducted in four municipalities in Sweden—Largetown, Smalltown, Midletown, and Subtown—that have desegregated some of their schools. These municipalities were chosen because they differ in size and their school markets have different characteristics. As shown in the review of previous research, school segregation relates to the school choice system and the proportion of independent schools. In some municipalities, there are competing independent schools (Largetown, Middletown, Subtown), while in others, public schools face no competition from independent schools (Smalltown).

The municipalities were also chosen for this study to include an example of private actors wanting to make a change by starting new schools in segregated areas, and in doing so, implementing a "new pedagogy" or concept of working for the schooling of children in ethnically segregated residential areas with poor social and cultural capital. The primary data-collection technique in this study was in-depth semi-structured interviews. In this paper, we are interested in how the process was steered in the different municipalities. Accordingly, we focus on the data concerning politicians, municipal school officials, and headmasters actively involved in the process of school desegregation.

The first stage of the data-collection process involved contacting the municipalities and identifying the key actors in the process of desegregating some of the schools. Who these key actors were differed between the municipalities; they were composed of a combination of local/municipal educational politicians, officials, and headmasters. In total, we interviewed 19 local school politicians, municipal school officials, and headmasters in four Swedish municipalities. The interviews were conducted between spring 2020 and spring 2021. Each interview lasted between about 60 and 90 minutes. Due to the COVID-19 pandemic, eight of the interviews were conducted via Zoom. The interviews focused on the background against which the process of desegregation was framed and its implementation from the perspectives of the different actors involved. All interviews were transcribed and anonymised.

For this paper, we focus on the data pertaining to the practices used to restructure the schools in the municipalities to deal with the "problem" of school segregation. We also want to understand the actors' reasoning and how a specific practice was adopted. The analysis of the interview data was conducted in two stages. The first stage involved reading the interviews to get a general understanding of the data we collected. Then, we read each interview separately to analyse and detect how the informants described their processes and the decisions made on how to desegregate the schools in the different municipalities, bringing the differences between the municipalities to light. The focus of the analysis for this paper was to identify and delineate how the process of desegregation was steered in the municipalities, i.e., to identify the steering practices adopted by the different municipalities to restructure the school composition to manage or counteract school segregation.

## 3. Desegregation of Hyper-Segregated Schools

An axiom in social studies says that how we construct and define a problem informs how we deal with this problem (see the Thomas dictum in sociology). In other words, how the problem of underperforming children in hyper-segregated schools is constructed informs how to manage the "problem". All the municipalities we examined defined the problem of academic underperformance of hyper-segregated schools in a similar way, and that is: lack of exposure of migrant children to the dominant group impairs their Swedish language development and ability, which, in turn, negatively "affects" their academic performance. This notion underlies the municipalities' understanding of the poor academic performance of hyper-segregated schools. Hence, to enhance the language skills of children with an immigrant background in hyper-segregated schools, the municipalities adopted different means of steering the social composition of the students.

### 3.1. Closing as a Practice to Change the Composition of School(s)

In all the municipalities, there had been a consensus for some time, particularly among the school officials, that the best option was to close a hyper-segregated school and distribute the students to different schools in the municipality:

> We have had discussions at the educational administrative level in the municipality about the poor performance of X school in Largetown for many years. Should we close the school? Provide the possibility for these children [children with migrant background] to meet other Swedish children whereby they can speak Swedish during their breaks and so on. (Official)

> In the last 10 years we have implemented major reforms, but these reforms and the measures we introduced did not improve the academic performance of the school. We have through different types of incentives and financial support from the government put a lot of money and resources into these schools. (Official)

However, one of the top school officials from Largetown noted that the idea was a political hot potato; the local municipal politicians had not been willing to change the status quo or do anything radical until recently. In spring 2019, the school director in Largetown decided once again to take up this issue with the ruling political majority in the municipality. According to the official, the school director was surprised that the ruling political leadership in the municipality welcomed the discussion:

> My boss once again took up the situation of the school with the chairperson of the school committee, and discussed certain ideas. I was surprised that she took that discussion, because I have been sitting in endless conversations with the politicians. There was no political interest before the spring of 2019. (Official)

> Unlike the school closing in Kaller X school was not characterised by violence, or vandalism. The school in Largetown had a very competent headmaster and teachers, and we have put a lot of money and other resources in the school, we could reduce the resources in this school by 25 per cent and it would still be the school with the highest resources in the municipality. There is nothing more we could do. Resources were not the problem or the answer, was the conclusion we reached, the problem is the nature of the composition of the pupils that made the teachers' work impossible. (Politician)

As evident above, this process to close the hyper-segregated school was initiated by a clique of school officials and politicians. The decision was based solely on the fact that the municipality had poured money and other resources into the school, but the school did not improve its academic performance. It is important to stress that the decision to close the school was not the first choice: it was the last resort. However, as will be evident later, the decision to close down the school in Largetown was not based on any commissioned study or internal study on how to deal with the "problem of school segregation" or improve the educational performance of the segregated school. The political administrative actors in the municipality instead made the decision based on their own conviction.

To avoid conflict, a public debate, and potential resistance by local school actors, the discussions and decisions to close the school were limited to a small number of school officials and politicians. As the outcome, they opted for the least controversial method and strategy to desegregate the school. They chose the same strategy they had used before to desegregate a similar hyper-segregated school in the municipality. The method they opted for then and now is to close the school and distribute the children to different schools using the school choice principle. The process of distribution was carried out centrally; the children were placed in different schools, and the only criterion in this process was to maintain a balance between migrant children and native Swedish students. The balance the municipality had to achieve in this process was to diversify the social composition of the schools without a school getting marked as an "immigrant school" and starting an avalanche of "White flight". In other words, the strategy of the municipality to counteract "White flight" was, according to one of the politicians, to ensure that:

> There was to be no school in Largetown that they could flee to. We strived for an equal distribution, a good balance between ethnically Swedish children and children of immigrant background. Largetown's private schools are popular, and to my knowledge, there is a good composition of children in Largetown's private schools. The English school in Largetown [a private school] is or has a segregation problem. It has more children with migrant background compared to ethnically Swedish children. (Politician)

One of the politicians we interviewed in Largetown stressed that school choice is a good mechanism to counteract segregation:

> School choice is a good instrument to offset school segregation, but it is important to acknowledge that the preconditions for choosing are not the same for everyone. If your mother tongue is not Swedish, you have no knowledge of the Swedish school system, have a poor school experience or background, it is not easy to choose. We have to understand that we cannot work with all the parents in the same way. When we talk to immigrant parents, we have to be very clear with information in their language, it costs a little bit of money, but it is necessary if the educational choice policy is to work. We . . . have, I think more than 96% of our pupils chose a school, and this is what is necessary for either the school choice system or the school choice to work for all. (Politician)

If one of these schools becomes too "popular" among children with an immigrant background, the central organisation uses the lottery system to control the social composition of children. As such, choice is not a given; the municipality regulates school placements to counteract tendencies of segregation of White or immigrant students in a specific school in the municipality. In Largetown, all students are provided with a bus card if there is a certain distance between the school and their home.

To sum up, the process and practice of school desegregation that Largetown adopted was the product of the political climate in the city. The decisions and discussions were kept within a specific group of politicians and school officials. The process and the decision to close the school was not transparent or based on any scientific study, and no alternative was discussed. The school officials had been wanting to close the school for a long time, but until recently, the political climate had not been conducive to the idea. Thus, when the opportunity arose, they tabled what they saw as the solution, which was to close the school, and the school politicians agreed. The school actors, including the headmaster and teachers, were informed only one day before the decision to close the school was made public. The parents and other actors were informed that the students in school X would be distributed to other schools based on the school choice practice. In other words, the different school actors were presented with a fait accompli; they were not given the opportunity to voice their opinion on the decision.

### 3.2. Opening an "Attractive School"

A second practice to steer the composition of students/children that we identified in the study was to open a new school in a stigmatised residential area. In such an instance, a foundation with a very high academic reputation in the local school market decided to open an independent school in a segregated area. The area is adjacent to one of the richest areas in a major Swedish city. The foundation runs one of the most prestigious independent schools in Sweden. Their ambition, according to one of the members of the foundation's executive board, is to improve the academic performance of schools in Sweden. They, furthermore, pointed out that opening a school in a segregated residential area is and should not be perceived as an "integration project":

> And I, I don't want to get stuck in this, that we're starting a school just to . . . just to contribute to integration, well because that's not it. Ehm, and then all the children, they, they become small players in this, it's like . . . They are people, and . . . everyone who wants to go to a good school and wants to find their way there, they should be able to go to the school of their choice. (Board member)

The decision to open a school in Subtown, according to the board members we interviewed, was made to give all children an equal education. The foundation does not perceive it as a social project of integration. The foundation chose this particular municipality because it has the reputation of being "a good school municipality":

> Yes, but, and then I can speak from my own perspective, because this municipality has a very good reputation as a school municipality and I think through the years

we have had teachers who speak well of the schools there, that's part of it all. But then, that this area is a diverse area [is] something that we in the foundation like to approach, we have actually been out there in other similar areas and all sorts of places to look at different possible premises and we . . . , but this came about because the negotiations and the premises, all fell into place. (Board member)

According to the officials of the foundation we interviewed, a better and more balanced composition of pupils could be achieved when strictly using the proximity principle and school choice policy/practice on their own, rather than combining both.

The logic, which seems to hold the consensus here, is to either use quotas or create a school with pupils from different socioeconomic backgrounds:

I just think it's so interesting that if you think that both geographical proximity and school choice have, ehm, very big disadvantages when they are used for . . . well just one or the other. But you get very . . . easily get rid of both disadvantages when you com . . . [clears her throat] combine them and then I think, like, why didn't the investigator think of that? Wh . . . why was it us two aunties who, who did it? Ehm, instead they come and say that you should group people into socioeconomic groups and, like, allocate quotas from that, that feels very forced you could say, when you have two criteria that very much are legally secure. (Board member)

The foundation's logic of starting a school in a segregated area is based on the belief that a good school with a good academic reputation will in time attract children from different socioeconomic backgrounds irrespective of the area in which the school is established or the backgrounds of its students. The foundation primarily targets children in the municipality using the proximity principle, while school choice attracts children from adjacent areas with high social, economic, and cultural capital, to create a social mix in the composition of the students:

. . . so that you know we've been open to kind of wanting to see if we can start a quality school, ehm, with tougher conditions kind of, or, well, a little bit more challenging. . . . I think like this, that if we have a really good school, then the children are going to like it, I mean, it's really fun to go to school when the school is good. And then, then they'll kind of, then I think it's not that important what the parents think to start with, they will also notice it. I think our thing is that we just have to create a really good school like we've envisioned. (Board member)

Another thing the foundation wanted to achieve by starting a school in a segregated residential area was to rehabilitate the reputation of the independent school, which, according to one of the board members we interviewed, had been tarnished:

Maybe it can also, ehm, increase credibility when you . . . after all, there are lots of people who have preconceived ideas about independent schools and such, so I suppose it's a little . . . if we were to succeed here, we would maybe be able to actually have more of an impact than if we kept to easier areas, you would think. (Board member)

To sum up, the choice of starting a school in a segregated area not only seems to have had a political agenda but also a social agenda: to make a difference. Opening a school with a good reputation and educational capital in a poor residential area was not simply a matter of providing quality education or making a philanthropic gesture, it was also a way to influence the (negative) discourse on independent school choice policy and practice. This practice, as implicit above, advantages children from certain socioeconomic strata—the middle and upper classes—and by starting an independent school in a stigmatised neighbourhood, the policy actors aimed to show that they take social responsibility.

*3.3. Bussing Children*

In Middletown, it was primarily the local school politicians who spearheaded the process of desegregating a school. The decision and the implementation were based on a study commissioned by the municipality and discussions with another municipality that had desegregated their school(s). Middletown chose to close a segregated school, but distribute the children based on the proximity principle. The students from the closed school were distributed to five other schools. According to the municipal school officials and politicians we interviewed, the decision to close the school was made because they had poured resources into this school, but to no avail; the school could not improve the children's academic performance. To provide an equal school experience for the children of immigrants, there were two options: build a new school or close the school. This is evident below:

> Clearly, it does not help to just provide the schools with more economic resources ... After all, we had, at the time [in 2018], really been in contact with, and heard a little about what the view was on these students with [lower] socioeconomic backgrounds and the importance of, of having a more heterogeneous mixture of students with different backgrounds. (Official)

> The way I see it, there are really two different ways to try to create equivalence. One is to build a new school and try to make this school attractive and strengthen the [segregated] area, and the other way is to close down and spread out the students to other parts of the city and other schools. So as to not create another segregated school, we're moving students to five different schools so that it, it won't be that many per school, and that's why we have made the decision that all are not moving to one place but to five different places. (Politician)

The pupils could choose another school, but only if the school had an open spot available and was not about to tip to become an "immigrant-dominated school".

The decision to take this approach was not transparent, having been made by a clique of politicians and educational officials. However, once the decision was made, the politicians and municipal school officials carried out a major information campaign to inform the different school stakeholders of the decision. This information process achieved a political and municipal school official consensus and unity regarding the decision to close the school:

> We from politics have been very active by meeting the schools, students, teachers, and parents. We have not just handed over an assignment to the administration and then came and sat at some table and made a decision on this. (Politician)

According to the informants we interviewed from Middletown, the decision to close the school was based on a study that the municipality conducted and internal discussions:

> Apart from the commissioned study, we created different types of workgroups and the headmasters were part of the process. We had a delegation that visited other municipalities to study their process, met the directors of the schools in the municipalities, the headmaster and researchers in one of the municipalities. We discussed their process of desegregation, the strategies they opted for, what worked and what did not work. ... We tried to understand, many things were discussed, but we wanted to come up with a recommendation that the politicians could accept. There were many reasonable ideas that were discussed, but politically impossible to implement. (Official)

The different working groups had a constant dialogue with each other and the political leadership in the municipality regarding the ideas or measures they discussed in this process, and they explored what was politically possible to implement. In other words, according to our informants, the idea was to recommend a measure that would garner the support of the main political parties.

*3.4. Merging as a Mechanism for Changing the Social Composition of the School(s)*

The desegregation of schools in Smalltown was initiated by a group of school principals. According to one of the participants in the group, the discussions began to pick up speed in 2011 and focused on how residential and school segregation negatively "affected" social cohesion and educational equity for children of an immigrant background. Similarly to Largetown or Middletown, the municipality had for many years provided three elementary schools (preschool–grade 6) with extra financial and pedagogical resources, but these measures did not improve the quality or the academic performance of the schools. One year before the municipality made the decision to desegregate the schools, 71% of the students in the lowest performing school an had immigrant background, only 18% of their parents had a university education, and just 45% of the students attained all knowledge credentials in year six. In the highest-performing school in the same municipality, 17% of the students had an immigrant background, 44% of their parents had university education, and more than 90% of the students attained the level of knowledge required in year six.

In 2017, the municipality merged three elementary schools for preschool to grade 6 (schools with children aged 6–12 years) into two schools for preschool to grade 3 (children aged 6–9 years) and one school for grades 4–6 (children aged 10–12 years). To achieve this, the officials in the municipality modified the school zoning and expanded the catchment areas to achieve a balanced social composition between ethnically Swedish children and children with an immigrant background in all schools in the municipality. The collective lobbying, voice of the principals, and perceived poor state of the academic performance of children of an immigrant background, with the support from school officials, together eventually made the politicians begin to understand the extent of the problem with the school organisation in the municipality and the stigma associated with some of its schools:

> It is so important to work to avoid stigma. So that some do not get [anything] and others do. Because everyone wants to fit in. That's what you're fighting for all your life. ( . . . ) So I think they trusted that all three of us principals wanted this. Not just the school that had the most problems. We pressed hard on the political committee. We will never be a better school municipality if we do nothing. (Official)

Discussions among principals focused on how school segregation generated problems at different levels in society in terms of both unequal school achievement and social cohesion. These discussions were passed onto officials and politicians.

Against this background, the policy actors in the municipality supposed that desegregation policies should begin at elementary school. They thought, or assumed, that the integration of children from different backgrounds at this level would improve school equity, counteract prejudice, and enhance social cohesion between different groups. In other words, they believed that younger children made new friends more easily than older children, and by breaking ethnic school segregation, the unknown would no longer be perceived as a threat:

> The more [people] who are my friends, the fewer [people] who are the others ( . . . ) Sometimes it has to do with skin colour and sometimes with something else. To categorise is a way to get order in one's life. But if the category boundaries go elsewhere, then we have done something good. (Principal)

The principals and officials revealed that during the processes in the different municipalities we studied, slightly different proposals were put forward for how schools could be organised in the municipalities to achieve school equity.

In this municipality, the politicians made the implementation of desegregation of the school dependent on using the existing schools: "There were different alternatives, but it ended up with two Preschool class–grade 3 schools and one 4–6 school in the 'urban area'. It was the most realistic when it comes to the size of the school buildings." According to the politicians we interviewed, the logical thing to do was to turn the previous immigrant-dense and lowest-performing school into a grade 4–6 school as its premises were best

adapted to receiving more students. Before the reorganisation, all schools were constructed to house 200–280 students. Today, the two P–3 schools have about 200 students each and the 4–6 school has just over 400 students. The longest distance between the schools is about 3 kilometres, which the policy actors believe facilitates students' easy transportation from different residential areas. The 6–9-year-old children are bussed to either of the two P–3 schools, and the 10–12-year-old children are biking/walking to a school in the immigrant-dense area. The bussing of students applies to both ethnically Swedish children and children with an immigrant background.

To create two equal P–3 primary schools, officials in the municipality changed the zoning of school catchment areas. The new school zone (catchment area) included residential areas that are dominated by different migrant ethnic groups. As such, the new zone was intended to achieve the social mix that school officials desired.

Thus, the steering and the implementation of desegregation policy in this municipality was made possible by collectively anchored involvement of various key people and continuous communication with the school personnel and parents. The politicians and school officials stressed that they had been discussing school segregation and inequality for a long time. During this period, the school committee's members (representing different political parties) did not changeover, meaning the same political alliances were in place as those in 2011:

> The three of us were in agreement . . . because it can be the case that when you are in opposition, you turn your coat after the wind and agree with those who whine and oppose. But I worked just as hard with this issue when I was in opposition as when I was chairman. It is a bit different how you are as a politician, I do not become a different person because I come into opposition ( . . . ) It is about having stubborn politicians. (Politician)

This stable constellation of the local school political actors was important not only for reaching a consensus on how they attributed the cause of school segregation, poor educational performance, and lack of social cohesion in the municipality. This understanding resulted in a consensus about how to steer the social composition of the schoolchildren in the municipality. This consensus is evident in the following statement by one of the local school politicians we interviewed: "No one was against it. Because what else are we going to do?" However, there was some resistance to the policy, particularly by the Centre Party and the Sweden Democrats. They argued that bussing young children between different areas and changing the organisation risked destroying well-functioning schools. The members in the local educational committee in the end had the political majority to implement the reorganisation of the schools to achieve the goal of school desegregation. In other words, in Smalltown, the political consensus was thin:

> In the end, I got a majority that thought we should do this. We even agreed that we from the Centre Party would vote differently on this issue. Therefore, I am sure that if the Sweden Democrats had requested a vote, my colleague also would have voted no. But now there was no voting, so he was snubbed, he never got that opportunity. (Politician)

In Smalltown, the local school politicians believed that a sustainable policy to combat school segregation in the short and long term required a political consensus among the main political establishments in the municipality. The politicians and school officials informed the parents and other actors why they made the decision to reorganise the school system in the municipality:

> So we organised a dialogue with our citizens, but we have also had one with our employees. There were those who agreed with the idea and believed in it. But then there were also those who did not believe in it, and they abandoned the ship. (Official)

To summarise, the different municipalities had similar understandings of the conundrum of residential and school segregation and agreed that a poor educational performance resulted from limited exposure of children with an immigrant background to the ethnically Swedish children's culture and society. This, in turn, negatively affected their Swedish language skills, educational performance, and career, and in the end, the social cohesion of Swedish society. The consensus shaped how the different municipalities approached the steering of the composition of the schools. In other words, although the municipalities constructed the cause and effect of school segregation in similar ways, they took slightly different approaches to steering and shaping the social composition of the schools. In our analysis, we identified the following strategies used by the municipalities to desegregate their schools: opening a new school in a segregated area, fusing existing schools, closing a hyper-segregated school, and distributing children to other municipal schools.

These practices of steering the composition of children/students in the municipalities came about as the result of a consensus among a network of critical actors who agreed on the construction of the problem of segregated schools and how to deal with or manage the problem. As such, steering the composition of children in a desired manner was contingent on the relationship between critical actors and the context. For a reform to be successful, it requires that structures of formal power and influential actors with individual agency (e.g., politicians and school officials) form a consensus on what the problem is and how to manage it; this is essential for determining the outcomes of a reform [41].

The results of the study show how different municipalities chose to change the composition of hyper-segregated schools: the "success" of this effort was contingent on the critical actors, particularly the local political actors' perceptions of what was viable, and how the politicians and local school officials defined "the problem of the poor performance of schools" with high rates of children of an immigrant background. However, politicians, school officials, headmasters, teachers, and parents can actively work against policies that might change the status quo, and parents can resist changes that they perceive will disadvantage the school experience of their children.

## 4. Discussion

In all the municipalities, a small group of school actors initiated the process and practice of desegregation of hyper-segregated schools. These groups consisted of local school politicians and school officials (the political administrative actors), with the exception of Smalltown where headmasters were also involved. In addition, in most of the municipalities, there was a common understanding between the different actors, particularly the political actors, irrespective of their ideological differences. These actors had worked together in different capacities for a long period.

Compared to the other municipalities, there was no political consensus in Largetown. This lack of consensus was evident in how the decision to desegregate was made and how the public and other school stakeholders were informed about the decision to close the school. It also informed the strategy they adopted. The decision was limited to a group of politicians and school officials, and the decision was made with no discussion with the relevant actors or different stakeholders. The strategy they used to desegregate the school followed the school choice principle. This method was intended to minimise potential conflicts within and between the different stakeholders in the municipality. In the other municipalities, as noted earlier, there was a political consensus among the political administrative actors; in their public blitz to inform different school actors, they showed a unity in the decision to reform the school system. The process of informing the different school stakeholders was not left to the school officials, as in Largetown.

In general, however, the actors that were involved in the processes in the different municipalities had a shared understanding of the "basic problem" with school inequality. They perceived and supposed that residential segregation was a major obstacle to achieving school integration and improving the academic performance of children with an immigrant background. In other words, it was the concentration of children with an immigrant

background and their lack of exposure to the dominant language and culture that were considered the "root cause" of the poor educational performance. In this context, it is important to stress that this perception is in line with research and international reports. For instance, as noted earlier, school segregation is linked to residential segregation and to the concentration of children with poor socioeconomic backgrounds in schools that are located in stigmatised residential areas. These areas and schools are populated by different ethnic groups that often are immigrants or of immigrant background [21–23]. From the perspective of steering, this is one of the basic and most critical conditions for success [38]. In other words, the political administrative actors in these municipalities had a common understanding of the problem of hyper-segregated schools.

In addition, the school officials and the politicians similarly perceived that school choice and proximity policies had exacerbated school segregation, as has been shown by Swedish and international research. However, as shown in our results, both of these principles were used and modified in different ways at the local level to create socially and ethnically mixed student groups. The municipalities applied these principles in three different ways: (i) combining proximity and school choice, (ii) using modified versions of proximity (as in Middletown and Smalltown), and (iii) using active school choice and a lottery system (as in Largetown).

The differences between the municipalities did not lie in how they defined and constructed the problem of the educational performance of hyper-segregated schools, but rather in the practices they used to desegregate the schools in their municipality. In the analysis of the data, we identified three types of steering practices used by the municipalities to desegregate the hyper-segregated schools: closing, opening, and fusion. An aspect that we would like to draw attention to is the stigmatisation of the different ethnoracial groups that dominate the so-called hyper-segregated areas in Sweden. These ethnoracialised groups are generally perceived as a "problem", and the hyper-segregated schools that their children are enrolled in are also perceived as problem schools. Thus, the root causes of school segregation are primarily residential; the areas in which hyper-segregated schools are located are stigmatised, and schools in these areas cannot attract White middle-class children. Furthermore, the policies of proximity and school choice present issues that the municipalities and many researchers have identified as creating school segregation. Accordingly, the long-term sustainability of the steering practices adopted by the municipalities can be questioned. For example, Largetown closed a hyper-segregated school some years ago, but problems still persist.

The peer effect was the primary assumption behind the policy of desegregation in the municipalities in this study. When the problem of school segregation and academic inequality is constructed in this way, it is reduced to the universal belief that exposure of immigrants to their Swedish peers will "positively impact" the Swedish language ability of children with migrant backgrounds, which, in turn, will improve their school performance. Although the peer effect was the main or explicit argument for desegregating the schools, we argue that a more thorough and systematic study of the problem would have led to the recommendation of other compensatory measures than closing the school. Furthermore, as many other studies have shown, desegregation alone cannot deal with other issues such as prejudice, racism, and discrimination. For instance, a study by Diamond et al. [42] showed that most White students perceived that Black students who were enrolled in advanced track programmes did not have the aptitude to attend. In other words, attending a desegregated school does not mean that minority children will adopt a pro-school disposition, given the attitude and assumptions of the teachers and their "White peers" about the academic ability of Black children [43].

To conclude, the municipalities' desegregation practices were not based on and did not depart from the cumulative research in the area, some of which we referred to here. Yet, the practice constituted a reduction of academic findings when applied to manage the composition of students in these schools. In other words, the municipalities disregarded a large body of research on school life and how alienating school practices lead working-class

children and minorities to adopt anti-school behaviours. In this context, any policy or practice that blindly stares at the social composition will risk failing in its attempts to solve the problems of inequality and lack of social cohesion.

**Author Contributions:** Conceptualization, A.O., S.L.; investigation; writing—original draft preparation, A.O., S.L. writing—review and editing. All authors have read and agreed to the published version of the manuscript.

**Funding:** This research was funded by Swedish Research Council, 2020-03355.

**Institutional Review Board Statement:** The study was conducted in accordance with the Declaration of Helsinki, and approved by the Etikprövningsmyndigheten, Sweden, Dnr 2021-00713, Approved: 25 March 2021.

**Informed Consent Statement:** Informed consent was obtained from all subjects involved in the study.

**Data Availability Statement:** Not applicable.

**Conflicts of Interest:** The authors declare no conflict of interest.

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
