# Peer review of "Local School Desegregation Practices in Sweden"

_education, doi:10.3390/educsci12080552_

Round 1

Reviewer 1 Report

This article discusses an important topic and presents some interesting data, but I would have liked to see a more developed, coherent and critical discussion running through the article. In order for the article to be accepted for publication, I found that it needs work in the following areas:

Proofreading: Throughout the article there are quite a lot of grammar mistakes, typos, empty spaces, and some sentences which are not clearly written. The paper could do with a careful proof-reading to eliminate these before resubmission.

Definitions and contextualisation: Several concepts/terms/ideas are used without sufficient definition or contextualisation – it would be good to go over the paper to identify areas that may not be obvious to an (international) reader and explain them. For example, I would like a definition of:

·     - what the authors mean by ‘hyper-segregated schools’ in the Swedish context. Are there any official definitions? What is considered hyper-segregated in one context may not be the same as in other contexts, and therefore a bit of detail would be useful.

·    - How is the concept ‘children with immigrant background’ defined or understood in Sweden? 

·       - What are P6, P3 and 4-6 schools?

·      - What is meant by independent schools and how big a proportion do they form of the Swedish school market?

The background sections move from Sweden to the US and other countries in Europe, but without much critical reflection on how their history and current social contexts differ. I think these sections would benefit from being rewritten to either 1) more clearly separate out the American and other national contexts and then say a few words about how they can/cannot be used to understand Swedish issues or 2) focus more specifically on the Swedish context and only briefly mention parallels to other countries. 

It would also be good with some more context of what is described on p. 10 as a ‘(negative) discourse on the free school choice policy and practice’. The Swedish discourse on free schools has not been mentioned before in the article and therefore needs explanation. Also, are free schools the same as independent schools in Sweden? On p. 3 it says that ‘“free schools” are similar to charter schools in the USA or independent schools in the UK. However, in the UK, free schools are also called free schools and independent schools are private and fee-paying.

On p. 13, the paper begins hinting at some of the political disagreements in Sweden over desegregation policies. If the authors want this to remain in the article, I think it should be more fully described, as an international reader will not necessarily know the Swedish political context.

Quotations: I am not sure the quotations that open the paper make much sense by themselves. I would recommend that they are saved for later, when you can include commentary on their significance. Some of the other quotes are also quite difficult to understand. I wonder if they could be shortened to make them more concise and leave more words for discussion.

Discussion section: This section could be more effectively used to critically analyse the views and quotes of the research participants in the study and more coherently discuss some of the possible alternative approaches hinted at in the abstract and in the last section of the discussion. 

On p. 15, the authors describe the contact hypothesis, focusing on its central idea that contact reduces prejudice. However, the research participants (and the paper in general) mainly discussed desegregation as a strategy to improve academic results. Although they may be linked, improving academic results and reducing prejudice are different objectives, and the authors would do well to clarify a bit more how they see the connection in general and in the analysis of their data.

The research participants quoted in the study seem to make an almost automatic link between immigrant status, low academic performance and low SES. This assumption needs to be more critically analysed and problematized. See for example Burgess (2014) Understanding the success of London’s schools, The Centre for Market and Public Organisation (CMPO), working paper 14/333, which argues that the success of London’s schools is due to their high immigrant composition. 

On p. 15, Race and Achievement Gap Theories are briefly mentioned, but not directly discussed in relation to the findings and therefore their relevance for the paper is a bit unclear. 

In the last section, the authors write that: ‘the municipalities disregarded a large body of research that examined school life, how the school life and school practices create alienating school practice which children working class and minorities to adapt anti school behaviours.’ The abstract also makes reference to 'multicultural education, anti-racist education etc, that could have led to develop different types of pedagogies that are relevant for this types of schools.' However, the paper itself includes very little reference to these bodies of research and nowhere is multicultural and anti-racist education described. In order to make these key points more clear and the argument more coherent, such theoretical and practical educational approaches should be included and discussed in more depth in the paper.   

Finally, I would be interested in knowing whether any changes have resulted from the restructuring/desegregation of schools in the four municipalities investigated. I realise that the authors may not have the data to explore this, but if they did, it would be an interesting addition.

Author Response

Please se the attached file

Reviewer 2 Report

Desegregation should be introduced for the keywords. In the end, qualitative analyzes should be categorized, for example:  views of politicians, directors, municipal officials and their attitude to desegregation.

It is important to briefly supplement the text with indications for educational practice that  take into account factors in the processes of the school desegregation.
